# Use of Virtual Environment and Virtual Prototypes in Co-Design: The Case of Hospital Design

**Tarja Tiainen [1] and Tiina Jouppila [2,*]**

[1]  Faculty of Information Technology and Communication, University of Tampere, 33014 Tampere, Finland; tarja.tiainen@tuni.fi

[2]  The Hospital District of South Ostrobothnia, 60220 Seinäjoki, Finland

[*]  Correspondence: tiina.jouppila@tuni.fi; Tel.: +358-50-474-5200

**Abstract:** Co-design is used for improving innovation, obtaining better solutions, and higher user satisfaction. In this paper we present how the use of a walk-in virtual environment and actual-size virtual prototypes support co-design. Unlike in most studies we presented the prototypes to users in an early phase of the design process. This study examines the co-design of healthcare facilities with multi-occupational groups. The practical case examines designing single-patient rooms for an intensive care unit. In this design process 238 participants of different hospital professions evaluated virtual prototypes in three iterative rounds. The participants improved the design by discussing their work practices. The virtual environment situation and actual size virtual prototypes make an easy environment for users to discuss and evaluate the design without any design knowledge. In addition to describing the co-design results we also outline some important issues and guidelines about creating the virtual prototypes and organizing the participants' visits in a virtual environment.

**Keywords:** co-design; virtual prototypes; virtual environment; hospital

---

## 1. Introduction

For decades there has been a focus on making employees' working environments better for them by including workers' work practice in the design process [1–4]. In order to succeed in this co-operation is needed between the professional designers and the workers who will work in the designed environment or with the designed products. When using co-design, the benefits are, for example, improved idea generation, better understanding of users' needs, and higher satisfaction of users [5]. In addition to long-lasting user participation in the design process (as in Scandinavian participatory development methods, see, e.g., [2,3]) users' involvement for a limited timeframe also gives a strong impact to the design process [6,7].

One problem in users' participation in the design process is users' difficulties in verbally describing their work know-how, as it is tacit knowledge about their work practice [8]. Furthermore, when the participants belong to different disciplines or professions, their underlying assumptions are different, which makes discussion difficult [9]. Another possible problem in co-design is that users could leave their own professional field and become more like designers [10–12]. Thus, they become less concerned with their own occupational needs.

Our study focuses on the multi-occupational co-design, which is an actual topic in co-design studies [13]. As our aim was to make it easy to participate in the process, the solution was to use virtual prototypes. When virtual prototypes are used, their evaluation does not require any special knowledge; instead, the participants can come there with their work knowledge and focus on their work tasks. This works especially well when the virtual prototypes are presented in their actual size,

as in a walk-in virtual environment (VE). In such a case, the users can try their actual work tasks within the designed objects (see, e.g., [8]).

The research question examines what kind of virtual prototype use supports multi-occupational co-design, more specifically, when a walk-in virtual reality solution is employed. In co-design it is important for the users and designers to be simultaneously present in the virtual prototypes since that supports their mutual understanding of the others' work tasks and practices [3,8,14,15]. When discussing the future work environment together, they can find a solution that fits everyone's needs.

The answer to the research question is sought with one actual hospital design case. In it a multi-occupational team evaluate and co-design a single patient room, bathroom and nurse station in an intensive care unit. This Finnish hospital design project was to design the first intensive and intermediate care unit based on a single-patient room concept. Evidence-based design supports the new trend of using single-patient rooms due to multiple benefits on outcomes on care, such as fewer hospital-acquired infections, fewer medical errors, and fewer patient transfers [16]. In addition to construction designers, different kinds of hospital professions also evaluated the design under process.

In our case the end-users participate in the design process only in commenting on tasks by the evaluation of virtual prototypes. By this solution we avoid the risk that is found in intensive co-design processes; in them the participative end-users become, at least partly, designers [12]. In order to avoid participants being expected to have designers' skills, such as understanding the two-dimensional design documents, we use virtual prototypes which are their actual size in a walk-in VE. This kind of immersive VE has advantages in design, such as better spatial perception [17], and it can affect the quality and duration of design reviews [18]. Furthermore, a 3D virtual prototype can be created in an early design phase and then it commits potential users to the design, and the requirements and needs of users can be obtained [19]. A walk-in VE is such an environment which works well for several people to participate simultaneously, so it provides a good opportunity for increasing understanding of others' work practice and their needs for the working environment. This is very important for designing an intensive care unit as various occupations work there with the patient and the environment needs to support all of that work. Our study provides a wider understanding of the possibilities VE gives for multi-occupational co-design.

In this article we first describe the scientific background, which includes both co-design and virtual prototypes. Then we describe our case, the co-design of intensive care rooms in a hospital building. Then we describe two benefits which were reached by co-design with virtual prototypes. These benefits are characterized based on the effect that the hospital staff participation had on the design process. However, we connect them to other researchers' findings. In addition to positive effects we also encountered some challenges within the process. We describe them by giving our solutions of how other co-design processes could avoid those possible problems. Finally, we end the article with a discussion of the results.

## 2. Scientific Background

We present two issues with respect to the scientific background. The first one describes what is meant with co-design. The second one presents tools for co-design, especially such virtual prototypes which can be used together in teams. The latter one is the main issue of this study.

### 2.1. Co-Design

The term co-creation is based on the idea that consumers are active players who are co-creators of value and co-developers of their own personalized experiences [20]. In addition to consumers, workers are also included in the development of their work tools and environment. This was done for decades based on work democracy, for example, in Scandinavia (e.g., [2,3]). Table 1 characterizes the main idea of user participation in the design process. There are a large number of benefits when co-design is used in the design process (see Table 2). The benefits can be observed based on the actor. The first object is how the co-design changes the design project. It improves the idea generation process and

improves the quality of innovations. The second object is the service user. For them co-design gives a better outcome of the design process, that being the most important issue in the design process. The third object is the organization, which can be either the designer organization or user organization. Co-design improves cooperation between the different groups of people. [6]

**Table 1.** Features of user participation in the design process (based on Kyng [21], p. 52).

| Element | Early Co-Design Process in the 1990s | Recent Co-Design |
|---|---|---|
| *Ideals* | Workplace democracy; Supporting user interest | User involvement; Better systems for all |
| *Users* | Workers as opposed to managers and owners | Non-wage earners, e.g., patients, customers |
| *Settings* | Workplaces | Non-workplaces, e.g., homes |
| *Techniques* | Experimental prototyping; amateur fieldwork | Experimental prototyping; professional fieldwork |

**Table 2.** Benefits of co-design in service design projects (based on Steen [5]).

| Benefits for … Theme | … the Service Design Project | … the Service Users | … the Organization |
|---|---|---|---|
| Improving idea generations | Better ideas Better knowledge about users' needs Better idea generations by bringing together different kind of users. | | Improving creativity Improving focus on users Better cooperation between different people and across disciplines |
| Improving the service | Higher quality of service definition More successful innovations | Higher quality of service | |
| Improving the project management | Better decision making Continuous improvements | | |
| Improving long term effects | | Higher satisfaction of users Educating users | More successful innovations Improved innovation practices Better public relations |

Although user participation in the design process has a long tradition, it varies in what is meant by user involvement in the design process. On one hand, users are considered as informants who can supply facts about work procedures but who have hardly any design knowledge and, therefore, should have little to say about particular design issues [10]. Users stay in their own competence area, and designers' task is to understand them and collect information for the design process [22]. Here this approach is labeled *Designers' move towards users* (Figure 1). On the other hand, there may be user representatives who participate for years in design projects and learn the design practice. Thus, users are expected to participate in the design process and know how designers think and work. Here, this approach is labeled *Users' move towards designers* (Figure 1). In that case, there is a risk that users become professional design experts and neglect the maintenance of their work expertise [10–12]. In addition to dividing the human-centered approaches to actor's roles (whose work is focused and who is asked to be flexible), they can be divided by their focus on either presenting the present situation (*what is*) or future situation (*what could be*) [22].

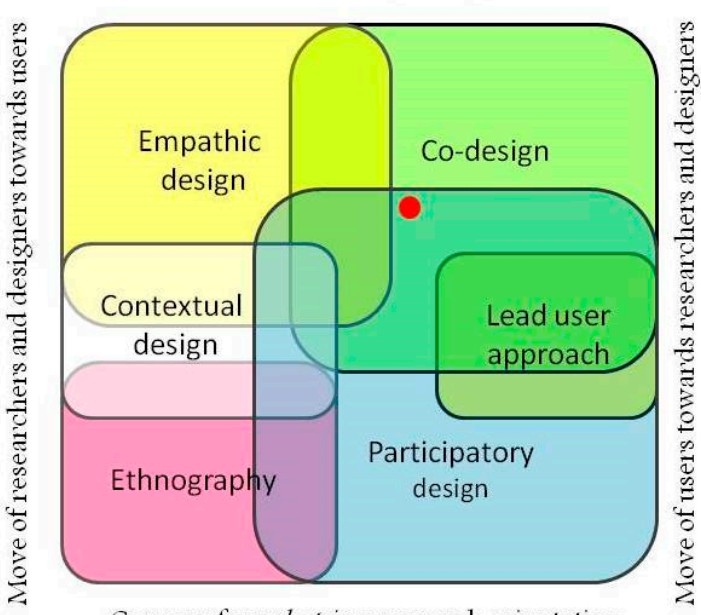

**Figure 1.** Different human-centered design approaches, with different starting points and emphases (based on Steen [22]). The red point in the figure presents where our study is located.

We can locate our study by these two aspects. In the first one, our study locates almost the middle, as the aim is that both designers and users can stay in their own occupational area. The second aspect focuses on the time dimension. In our case, the bases for users' evaluation of the prototypes is the present situation. However, the design focuses on the future situation, i.e., working in the new building, so the main issue in the evaluation of the virtual prototypes is the designed new situation.

Figure 1 describes the alternative user-centered methods. Participatory design and co-design are partly located over each other. In our study we have used the fundamental aspects of participatory design [4], as our study focuses on worker situation and the target is mutual learning between users and designers. However, we also utilize the features of co-design, as the idea of user involvement and better situation for all, and the use of experimental prototyping [21]. The location of our study is marked with a red point in Figure 1.

*2.2. Co-Design with Virtual Prototypes*

There is a variation of methods and tools which support co-design. A common and successful method is storytelling, as it is how people communicate in their everyday life and it does not require any new tools [23,24]. In addition to verbal description storytelling can use visual elements. In this study we went further towards the visual direction: the use of virtual prototypes so that participants can show their work practices to others.

Virtual prototypes offer a new platform for discussing prospective use and possible improvements. Virtual prototypes include using computers to visualize the prototype, but there is variation on how realistic the visual image is. In addition to visual images, virtual prototypes might also include some functions of the product. The benefit of using virtual prototypes is that they reveal more developing ideas than physical prototypes [25]. Additionally, immersive virtual prototypes have the potential to increase the quality of reviews and decrease the cost and duration of the design process [18]. Furthermore, virtual prototypes are easy to modify so new ideas can be quickly incorporated and new versions can be taken under evaluation [25].

There is a wide variety of virtual prototypes and several methods of using them in co-design. They are connected to each other: technology provides possibilities for co-design and the type of virtual

prototype affects co-design. One end is the use of screen solutions in presenting virtual prototypes, such as avatars in SecondLife [26], whereas the other end is to connect the prototype to a simulation model, such as designing workplace ergonomics with simulated virtual prototypes [27]. Furthermore, virtual prototypes can be presented in a walk-in VE. In such a case, users can try their work tasks with the actual size virtual prototypes (e.g., [25]).

When focusing on the use of virtual prototypes in co-design two issues are important. First one is the size in which the prototypes are presented. If the aim is that the users can try how they could act within a designed environment, the virtual prototypes need to be in actual size. However, if a screen solution, such as SecondLife, is used, then the virtual prototypes are small, and it is only possible to imagine how the work tasks could be done. Instead, actual-size prototypes are presented in walk-in VEs [28], as in the Cave, and in augmented reality, as with HoloLens.

The second issue is how the virtual prototype setting works for multiple people, which is needed in a co-design situation. When the aim in co-design is that mutual communication in the organization gets better, it is required that during the co-design process several people can be simultaneously in the same room evaluating the virtual prototypes and presenting to others how they would perform their work tasks within the prototypes. For such needs only a walk-in VE works, as the participants see the same image (i.e., the same virtual prototypes) and each other. Thus, they see what others are doing but also their nonverbal signs (gestures and expressions) during the co-design session. That is not possible with most augmented reality lenses; for example, HoloLens is just for one person.

In their review Kim et al. [13] classified 150 journal papers (2005–2011) in the field of the built environment concerning the use of virtual reality and virtual environment. Collaborative use of VR systems was used in 36 articles (26%). Kim et al. also predicted that multidisciplinary collaboration will become a standard in built environments [13].

When using virtual prototypes, the level of fidelity for prototypes needs to be decided. High-fidelity prototypes look like final products with a large amount of detail, whereas low-fidelity prototypes are insufficient, like sketches and virtual prototypes without texture or other details, for example. There are opposite views as to what level of fidelity works best in user evaluation. On the one hand, high-fidelity prototypes help professionals, and even non-professional participants, perform accurate design reviews [18]. On the other hand, when low-fidelity prototypes are used, the non-technical users have more space to evaluate and develop the presented prototypes [25,29]. In our case we used both high- and low-fidelity prototypes.

## 3. Research Methods and the Case of Hospital Design

The research method of this study is action research. The practical problem, which we attempted to solve, is the design of a new intensive and intermediate care unit. The theoretical knowledge that is utilized in solving the practical problem is co-design and virtual prototypes. For the practical case, this theoretical basis calls for nurses and other hospital workers' presence during the design process. In addition to using the theoretical knowledge in solving the practical problem, practical problem-solving is used to attain theoretical knowledge.

In the health care sector, it is realized that workers', i.e., nurses and doctors, participation in the design process is important for supporting innovations (e.g., [30]), since new co-design languages to facilitate cross-cultural communication are valued [11]. Our study utilizes these positive effects on co-design within the health care sector.

This study was carried out within a hospital design and construction project (2014–2018) in Seinäjoki, Finland. This hospital was the first hospital in Finland with single-patient rooms in intensive and intermediate care units. Single-patient rooms were designed since studies prove that this solution provides several benefits, such as fewer hospital-acquired infections, fewer medical errors, improved patient sleep, improved patient privacy and confidentiality, and fewer patient transfers [16]. The design project utilized multiple tools to support the hospital staff's participation in the design project.

The single patient room concept will change the work and work processes [31]. The intensive care room, the intermediate care room, bathroom, and nurse station were chosen to be designed. These rooms were chosen to co-design with the hospital staff, because of the number and the significance of the rooms. The total number of patient rooms was 24 and they formed the core of the new unit. The size and the form of the rooms, as well as their mutual situation, needed to be ensured for the changeover to single-patient rooms. Co-design virtual prototypes were chosen as the methods and the tools to support the design process. The hospital staff's role in the co-design process was to evaluate and comment on the draft which the architect team had designed. By using actual-size virtual prototypes, the hospital staff could try how they would work in the designed environment. Additionally, the idea of co-design was that multiple staff members participated simultaneously as they could discuss several work processes that need space and equipment around the patient bed. The hospital staff members were supported by their colleagues which made their comments more powerful.

In this section of the research methods, we first present what kind of VE we used in presenting the virtual prototypes. Second, we outline the process of presenting and evaluating virtual prototypes. Third, we present what kind of users visited the VE and evaluated the prototypes. Finally, we outline how the data was analyzed.

### 3.1. Walk-In VE

For making simultaneous participation possible the virtual prototypes were presented in a walk-in VE. It is a room comprised of three walls, a ceiling, and a floor. The fourth wall was open in order to allow access into the space (Figure 2). The measures of the walls are $3.0 \times 2.5$ meters each. Images generated using computer graphics cards were projected onto these surfaces, which, when viewed through stereoscopic glasses, were transformed into a 3D full-scale environment. The most important property of a walk-in VE is its scale, i.e., the ability of visitors to perceive the environment as almost real. Visitors can move to some extent within the space itself and travel longer distances (that means seeing other parts of the building) with the help of a 3D mouse.

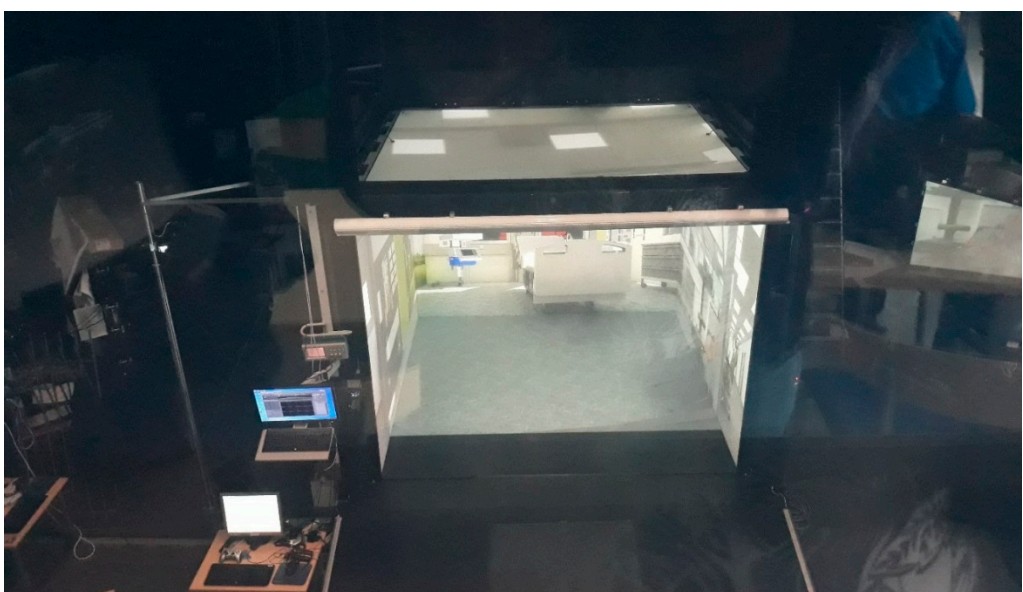

**Figure 2.** The walk-in VE is situated in a bigger room. There are projectors for each wall.

The software used in this project was TreeC Technology's VR4MAX. The virtual prototypes were created by the staff of virtual reality laboratory of Seinäjoki University of Applied Sciences, Finland. This was also the venue in which the prototypes were presented and evaluated. To help the navigation in the model the virtual prototypes were fixed on the floor level, so flying was impossible. Additionally, going through walls was restricted, so it was impossible to get out of the virtual prototype. The third

prototype included some functionality, such as sliding doors and the possibility to move medical ceiling-mounted joints.

*3.2. Presenting the Virtual Prototypes*

The designing of the new hospital unit required virtual prototypes, which included the two single-patient rooms, bathroom, and nurse station. The final version of the unit plan was created in three rounds of virtual prototypes. Between the rounds the new prototypes were modified based on the users' discussions of the earlier one.

The precision and fidelity of virtual prototypes increased round by round. The first and second virtual prototypes included only the patient bed and ceiling-mounted booms. The requirements for room size and shape, and furniture and equipment placement and quality were attempted to be discovered with these virtual prototypes from participants. Low-fidelity virtual prototypes produced discussions of missing furniture and equipment and their number and placement. The participants evaluated the placement of furniture and the shape of the rooms relating to the visibility of the patients. Additionally, the absence of windows in these earlier virtual prototypes caused a discussion of the importance of natural light and views of nature.

The third virtual prototype included all needed furniture, equipment, and accessories. The virtual prototype contained a window and a landscape painting to add comfort. The feedback of this last virtual prototype from participants focused on placement and quality of furniture, equipment and accessories.

The VE visit started with presenting information about the project, the location of the VE, the duration and the objects of the visits. The participants were informed that the rooms were incomplete, so their comments were needed for their future design. Additionally, information pertaining to videotaping, photographing, and recording their visits was provided. Before the actual design session, the researcher introduced the project group, demonstrated how the work would happen in practice in the VE, and explained outlines for the virtual prototype presentation. Participants were told that they could interrupt and comment anytime during the presentation.

The project group had planned the route for the virtual prototype presentation. That included a visit in an intensive care room, a bathroom, a nurse station, and then into another intermediate care room and a bathroom. The architect presented spaces with the same description to all the groups. The planned route took approximately 15 min. After that groups could return to the virtual rooms they wanted to evaluate more closely. All the time the participants could interrupt and ask for details or comment. The visit in the virtual rooms lasted between 30 and 40 min.

The setting of a VE visit is presented in Figure 3. The distribution of assignments in project group was the following:

- *Architect* presented the virtual rooms and answered comments on the technical and design side.
- *Navigator* guided in the viewing of the rooms and used the 3D mouse.
- *Researcher* opened the conversation if needed, asked for details and grounds for changes, gathered the comments and change requests.
- *Project worker* answered comments on the functional side.
- *Photographer* took videos and photos.

The evaluation of the rooms was completed in three rounds. The researcher gathered the comments during the VE visits. The project group discussed the change propositions and evaluated them. The ones that were evaluated to increase the prototypes or which had substantial reasoning, were taken for further design.

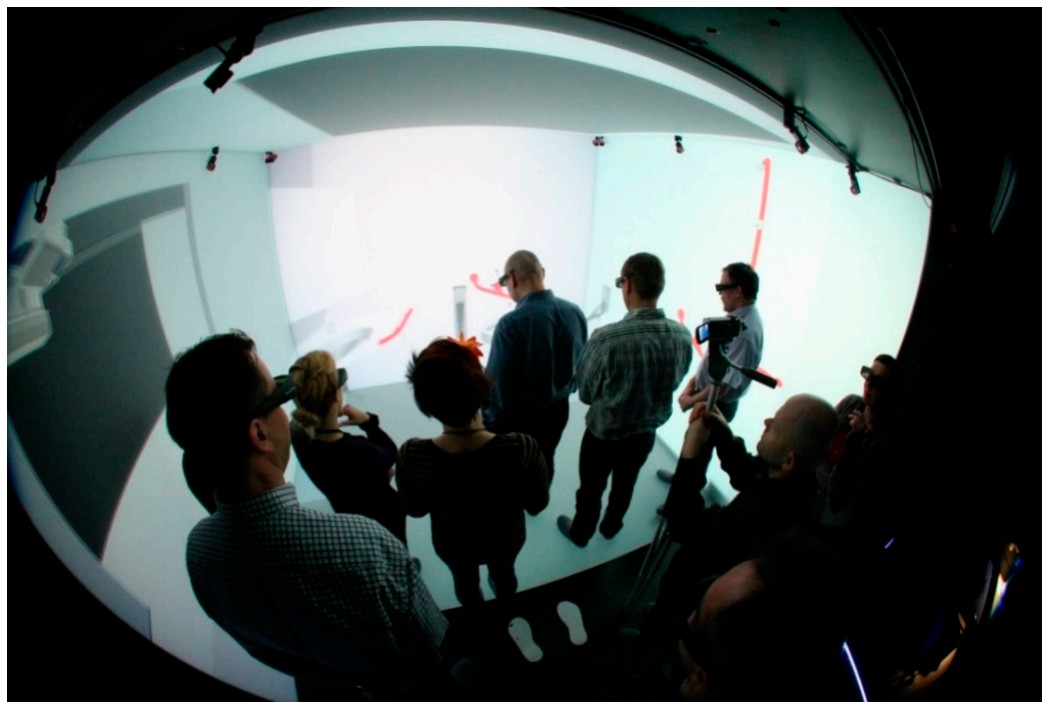

**Figure 3.** The setting of a VE visit.

In addition to mutual discussion the participants were asked to complete a questionnaire in which they could analyze the properties of the spaces more systematically as well as evaluate the virtual environment experience. The aim of the questionnaire was to make sure that all comments and evaluation would be brought together. For this study only a part of the questions was analyzed. That part included statements which dealt with two issues: first, how good the co-design in VE was, and second, what kind of feeling the VE gave to the participants.

### 3.3. Users Who Visited the VE

The VE visits took place in the spring of 2015. The timetable and number of groups and participants are shown in Table 3. There were 47 multi-occupational groups altogether. There were approximately 4–6 participants in each group. In one group there was only one participant and in few groups more than six participants. The total number of visitors was 238.

**Table 3.** The timetable and number of groups and participants.

| Dates | March 2015 | April 2015 | May 2015 | All |
|---|---|---|---|---|
| Groups | 14 | 18 | 15 | 47 |
| Participants | 61 | 89 | 88 | 238 |

Most of the participants (157) visited the VE only once, whereas one third of the participants (81) attended two or more rounds. The largest occupational group was nurses: 67% of participants. The rest were physicians, designers, cleaners, technicians, health and facility managers, students, and other healthcare professionals. Most of the participants were women, 79% in all. The median age was 45 years ranging from 24–65 years, calculated from those participants who disclosed their age. However, 26% did not divulge their age. The work experience ranged from 1–40 years and the median was 19 years.

The groups varied by occupation, but there were one or more nurses in all groups. Different occupational groups highlighted the requirements of their own profession and work set for the spaces.

The requirements were partly similar, but partly contradicted each other. There were some participants from other hospitals, too, and this brought into discussion differences between hospital cultures.

In addition to discussion of virtual prototypes there were discussions of visibility, work processes, and the number of nurses. The method of working was changing due to single-patient rooms. Additionally, the conversations in the VE included issues that cannot be evaluated in VE but are very important for design, like floor materials.

*3.4. The Analysis of the Gathered Data*

Two kinds of data were used for this analysis. The first one was a questionnaire, which the participants answered just after their VE visit. This data was analyzed statistically. The questionnaire included seven statements about the functionality of the groups' co-design within the VE visits and the feeling in being in the VE. It was calculated how many participants agreed to the statement. This data outlines participants' (i.e., hospital staff's) opinion of the co-design with virtual prototypes.

The second one is the discussion during the VE visit and the development of virtual prototypes. These data are notes which the researcher made based on her observation of the VE visits. Although she made them during the actual VE visit, they were also videotaped and it was possible to check from the video what actually happened.

## 4. Results

The results consist of three issues. The first one outlines the general picture of participants' views of their visits in the VE and their participation in co-design. This part is based on the analysis of the questionnaire. The second issue focuses on the benefit of organizing co-design sessions with virtual prototypes. The result is that hospital staff's work practice become visible to designer team and with it they can create better designs. The third issue is the benefit that hospital staff discussed the risks which the new design might include, and they co-operatively found solutions to overcome the risks. In the next sections we present the three issues with examples from the empirical data.

*4.1. General Picture: Useful Participation*

Just after the VE visit, the participants filled out the questionnaire about the VE and the co-design process. The results show that they found the co-design very useful, as presented in Figure 4. The questionnaire included seven statements, which dealt with the functionality of the co-design with virtual prototypes. The statements were evaluated by how much the participants agreed with them. This was done with a five-point scale from "totally agreed" to "totally disagreed". Figure 4 includes the participants' evaluations of the statements after all three VE visit rounds. The evaluations of the rounds are presented together, although the virtual prototypes and their fidelity varied.

The questionnaire included five statements, which dealt with the following:

- Participation was useful (totally and partly agreed 96%);
- Designing in groups was a useful method (totally and partly agreed 95%);
- In discussion, my opinion was taken into account (totally and partly agreed 85%);
- Commenting was easy (totally and partly agreed 92%); and
- Active participation was easy (totally and partly agreed 89%).

Additionally, the questionnaire included two statements, which focused on the immersion of the VE. In the evaluation they were also quite highly agreed:

- I felt like being inside virtual rooms (totally and partly agreed 96%); and
- Space was real and natural (totally and partly agreed 94%).

The participants' general feeling was that participation in co-design with virtual prototypes was useful. This evaluation is done based on the answers in the questionnaire.

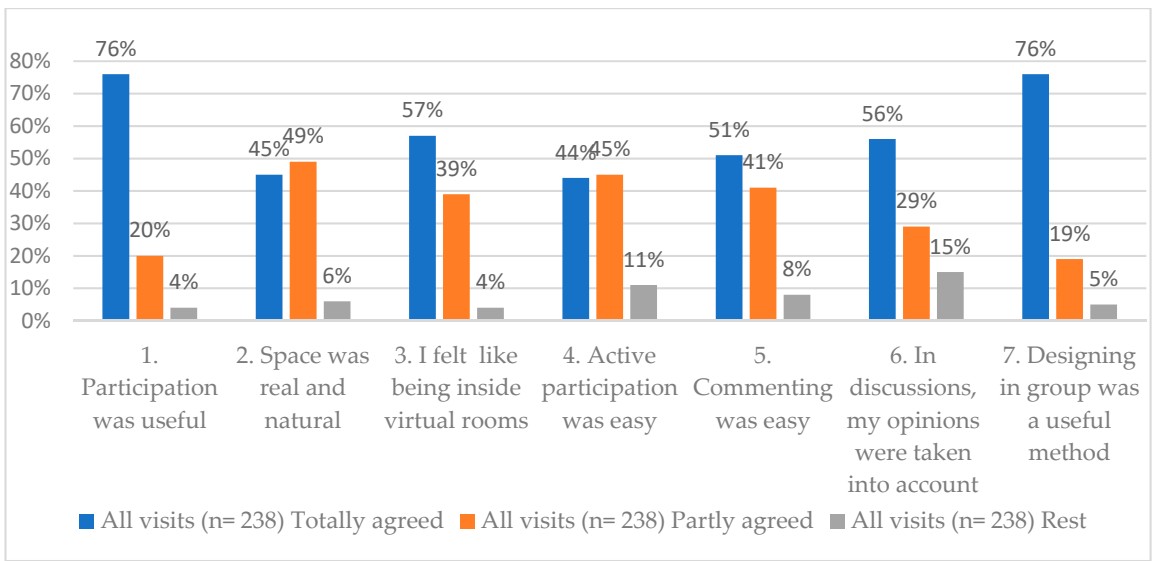

**Figure 4.** Results of the questionnaire evaluating co-design and virtual environment. ("Rest" includes answers "don't know", "partly disagreed", "totally disagreed", and missing cases).

### 4.2. Benefit 1: Professionals' Work Practice to Designer Team

The benefit of virtual prototypes is to support communication within the design teams and the users of the design outcome [29]. Especially when low-fidelity prototypes are used, the non-technical users have more space to evaluate and develop the presented prototypes [25,29]. The use of virtual prototypes gives hospital staff a possibility to stay in their own professional field without trying to act like professional designers [8].

In our hospital design case, the target was to support hospital staff's participation by discussion and finding a good solution that fits every situation in which the room will be used. The idea was that the actual size virtual prototype gives a possibility to discuss together how alternative professional groups will work there. The first version of the prototype for hospital staff to evaluate is presented in Figure 5. The prototype presents a part of an intensive care unit which includes a nurse station for two nurses and two-single-patient rooms which both have their own bathroom. One issue that was used for improving the discussion was that, in the first evaluation round, the presented virtual prototypes were somehow low-fidelity. The prototypes were missing many detailed objects and wall textures. The aim of getting the participants to discuss together was fulfilled excellently in our case. In the VE visits the participants brought up many issues that reflected their concerns for future facilities.

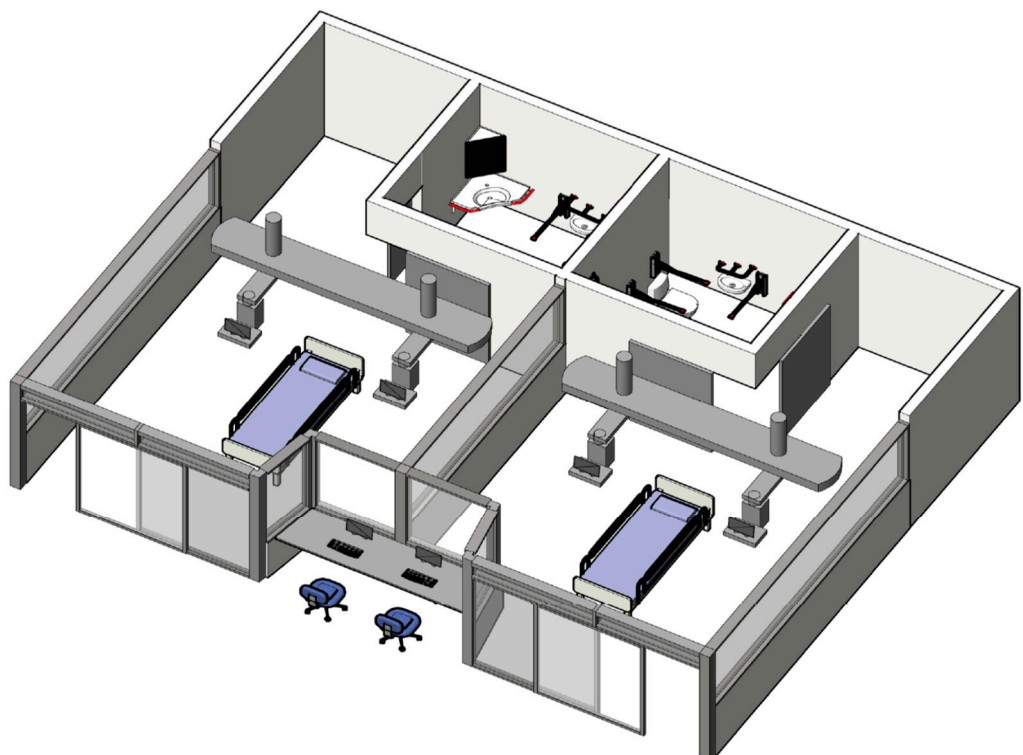

**Figure 5.** The first prototype.

The first issue was the size of the rooms. The virtual prototypes appear to be adequate in size but whether this proves to be the case in a work situation depends on how full of equipment and staff the room is in a critical situation. The participants listed the missing furniture and equipment and presented their position related to the patient bed. Additionally, participants listed other items needed in the room like a sink, storage units, armchairs for relatives, and accessories. Open floor space around the patient bed is very important. Ceiling-mounted booms for respirators and other equipment and ceiling mounted patient lift and movable objects were evaluated to be practical by sparing floor space and avoiding wires on the floor.

The second critical issue in patient care is the transportation of a patient with all the equipment. The doors should be wide enough. Usually many staff members participate in the process. From that point of view the location of the bathroom was very critical. First, a patient is moved from the bed to a stretcher or a wheelchair with all the necessary equipment and then transported to the bathroom. The route should be accessible and clear. The approved location for the bathroom was found in the third virtual prototype (Figure 6.).

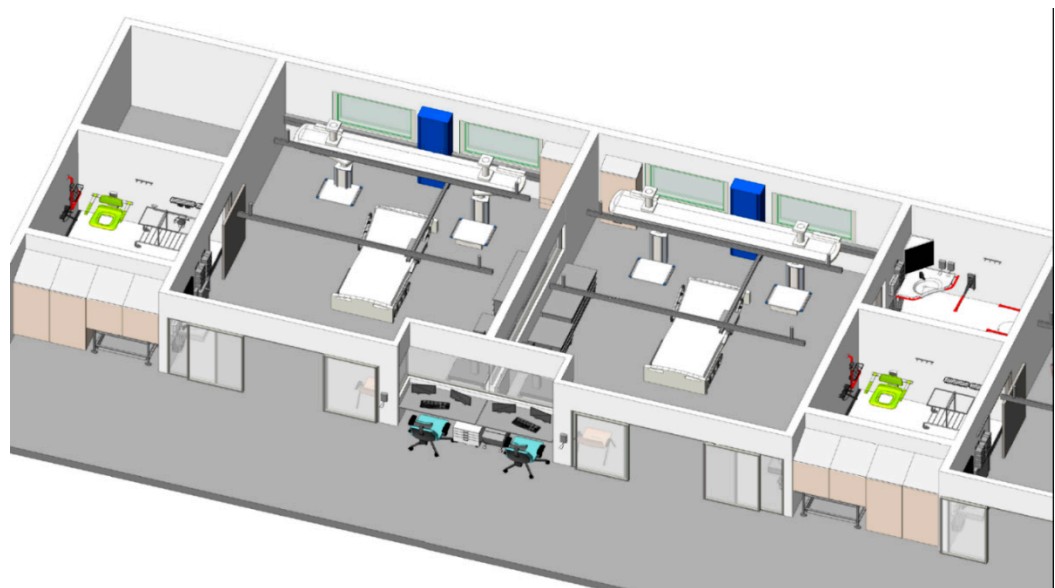

**Figure 6.** The third prototype.

The third issue was that it was easy to comment on the virtual prototypes for participants. All the participants had the same view and it was possible to show the item that needed to be changed. Nurses and other hospital professionals could use their professional language describing their work processes and space requirements. It was easy to find shortcomings as well. As the groups were multi-occupational the different requirements concerning the rooms were discovered and discussed.

The fourth issue discussed was the request for the design of the whole new unit with virtual prototypes. It is important to see how different rooms are situated in relation to each other and the distances between the rooms. During a work shift the staff must go to a storage room, a utility room, a pharmacy room, a restroom and nurses' office. The layout of the unit should be designed so that it saves the steps taken during a shift. It was impossible to design the whole unit during the project, but all the ideas about layout were driven forward.

### 4.3. Benefit 2: Discussions of Risks and the Solutions

In co-design the benefits for users are getting higher quality design outcomes and, in the longer term, higher satisfaction [5]. For reaching these goals one part is to discuss and solve the possible problems which users see when they evaluate the design. In our case the multi-occupational teams discussed the possible risks when the virtual prototypes were observed and evaluated. In these situations, two possible risks were commonly discussed. The first of them is how the nurses get assistance, and the second one is the limited quantity of natural light in the rooms.

The changeover to single-patient rooms was the greatest challenge in the design process, because it causes changes in work processes which also highlight the risks in care to be handled. The old unit was multi-bedded. Patients were separated by curtains, so they were near each other. It was easy for nurses to ask for assistance from other nurses. The new unit with single-patient rooms highlighted three risks to solve. They are assistance, visibility, and audibility. The nurses were concerned how to get assistance when they are working in different rooms. They were concerned that they cannot see each other and the patients and hear the patients. Similar concerns can be found in earlier studies of transitions to single-patient rooms [31].

In our study virtual prototypes were used for being sure that it was easy to get assistance by having visibility between the rooms. The visibility was needed for nurses to see their patients and other nurses. Audibility was also needed, so that nurses could hear what was happening, e.g., how the patient was breathing or if other nurses needed assistance. The solution was studied with virtual prototypes. The walls and sliding doors towards the corridor were made of glass providing visibility.

Additionally, the nurse station between two patient rooms provides good visibility to these two rooms. However, privacy was also needed, so that the patients could not see what was done to other patients or even who the other patients were. The minimum height of the window between two patient rooms was measured so that a patient lying in the bed could not see to the other room.

In discussion about the risks the second issue was windows and natural light in patient rooms. Studies prove that natural light supports patients' healing [16]. During the design occurred that some patient rooms are not provided with windows. The proposed solution was to design the layout so that most of the patient rooms will have windows and skylight windows in the corridors.

In our case of designing a hospital intensive care unit, the hospital staff discussed their concerns for when the unit planned to change from multi patient rooms to single-patient rooms. The discussion happened in a walk-in VE with virtual prototypes in multi-professional teams. The hospital staff's concerns were taken seriously, and the solutions were found.

## 5. Practical Guidelines for Organizing VE visits

In our study we used a walk-in VE for multi-occupational co-design. The hospital staff participated in the design process by evaluating the virtual prototypes and improving the design. During the co-design a high number of VE visits with 238 participants were organized in a short period. The three rounds of VE visits and development of the virtual prototypes were carried out in three months. As the timetable was tight and the number of participants high, we needed to plan the VE visits carefully. Within organizing the VE visits we made practical findings and we want to share our knowledge to others who use virtual prototypes in co-design.

*Guideline 1: Prototype.* The participants expressed the wish to design the whole area. In larger layouts a bird's eye view is helpful to comprehend the whole design, although it increases the risk for simulation sickness [32]. In our case the virtual prototypes were modelled so that the floor level was fixed (flying was impossible); it was also impossible to get out of the prototype. Still, some of the participants had dizziness and inconvenience in balance and they did not like the visit due to those symptoms. These problems occurred when using automated navigation in which the layout was presented with a route that was decided beforehand. As it caused problems to some participants, the speed of the tour was too high.

In addition to the whole landscape the outfit of the prototypes under evaluation needs to be decided. An important point is the level of fidelity, since it affects the users' evaluation [25,33]. In our case, the first prototypes were low-fidelity and the participants then focused on the missing elements of the prototype. The last prototype was higher fidelity and then the focus was on the details, such as the placement and type of furniture and accessories. As de Casenave and Lugo [18] claim in their article, high-fidelity prototypes help professionals, and even non-professional participants, to perform accurate design reviews. Furthermore, with this prototype less improvements were presented.

*Guideline 2: Improving the prototype.* In using virtual prototypes in co-design, the idea is that the prototypes are improved between the evaluation rounds. The first step is to analyze the users' comments and improvement ideas and then decide which ones will be taken to the next evaluation round. After that is the implementation task, which depends on the used VE environment and the implementation tools how quickly the changes are possible to be made. In our case, some weeks between the evaluation rounds was too short a time and there was not enough time to test the improved prototype before the next VE visit. The final task before new VE visits is testing the new virtual prototype by the project group.

*Guideline 3: Groups and rounds.* One of the main benefits of co-design is to increase the mutual understanding of workplace work [5]. To reach that, multi-occupational groups need to be organized. In our case the groups were formed based on the staff's working timetable and ensured that several different occupational members participated in each group. Another solution could be on the bases of work processes. The other issue about groups that must be decided is how many evaluation rounds the same persons take part in. In our case, most participants participated only once. The participants

who participated two or three times commented that it was easier to concentrate on evaluating the virtual prototypes when the VE was familiar. However, if the same persons participate in too many evaluation rounds, it is possible that they become more like designers [12].

*Guideline 4: Navigation*. The navigation arrangements should be considered carefully, since some VE visitors like to use the navigation tool (3D tool), whereas others avoid using it, and that does not depend on how difficult they find the use of a 3D tool [34]. It is recommended that the navigation task is given to a supported person, not to anyone from the evaluation group [35]. In our case, in most of the VE visits the navigation was done by an experienced navigator and that would have been a good solution for all the VE visits.

*Guideline 5: Duration of the VE visit.* When deciding the duration of the VE visit two kinds of issues need to be considered. For the participants the VE visit is a difficult task that demands concentration, so the visit must not be too long. A VE visit with new technology requires learning from the participants. Furthermore, the evaluation task done in a multi-occupational group is also a challenging task. We used approximately 30–40 min per group in VE.

The other issue that needs to be considered is the cost of the VE visits. The visits take many working hours. In addition to the work of the design project team the participants use their working time for the evaluation task. In our case that part was almost 300 h, as there was some distance from their workplace to the VE laboratory. The other part of the costs is the use of the VE laboratory. However, the costs of VE are coming down as more technological options come to market.

## 6. Discussion

We presented an action research study in which the practical part is a real hospital design case. In our case the design target was to move from a multi-bedded intensive care unit to single-patient rooms. The new wing to the hospital building was under design but also the nurses' work practice was changing. Single-patient rooms are known to have many benefits, however, there were also many risks [16,31]. When the design plan was under multi-occupational evaluation with using virtual prototypes, the possible risks were discussed and solutions were found. The participants evaluated the VE visits and valued co-design high, as presented in Figure 4. The design was done in 2015, then the new building as built and taken into use in 2018. The result will be evaluated in the autumn of 2019.

Our study outlines co-design, which Steen [22] localizes to human-centered design approaches, as shown in Figure 1. The location of the approaches is presented by two dimensions, which are:

1. The dimension of whose profession is dominating (designers or users); and
2. The time dimension from present situation (what is) to future possibilities (what could be).

Our study was located quite in the middle, just in the time dimension and a bit more to the future. Furthermore, Steen's figure includes alternative methods of human-centered design approaches [22]. Our study belongs to co-design, but also takes elements from participatory design. The concept of co-design is more focused on consumers' role on design [20], whereas our case focuses on workers' participation, as is done in participatory design [3,4].

Our study proves that discussion between designers and multi-occupational users can happen in a neutral space, in which neither profession dominates. There is a large risk in co-design that users become designers if the co-design process is long and the method requires design knowledge [12]. We created a neutral space for discussion about the design with virtual prototypes which are presented in a walk-in VE. In such a situation users do not need to understand design methods or tools; instead they can focus on their work tasks and present them to designers, who can include those needs in the design process.

When virtual prototypes are used one issue that needs to be solved is the level of fidelity of the prototypes. Some studies (e.g., [18]) claim that the higher the fidelity level, the easier it is for users to evaluate the prototype. Some other studies (e.g., [25,29]) claim that lower fidelity gives more freedom to users to evaluate the prototype, as the high-fidelity prototypes might seem to users that they are

ready designs, not under development anymore. In our study we used both high- and low-fidelity prototypes. In our case, during the first two VE rounds the virtual prototypes were of lower fidelity, e.g., the texture was missing, whereas in the third VE round the virtual prototypes were of higher fidelity. In our case, both the high- and low-fidelity virtual prototypes supported the mutual discussion in the VE visits.

**Author Contributions:** T.T.: theoretical background and supervision; T.J.: participating and analyzing the practical case; and T.T. and T.J.: wrote the paper together.

**Funding:** This research was funded by Tekes.

**Acknowledgments:** Jutta-Noora Kuosa for improving the language.

**Conflicts of Interest:** The authors declare no conflict of interest. The funders had no role in the design of the study; in the collection, analyses, or interpretation of data; in the writing of the manuscript, or in the decision to publish the results.

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
