# Peer review of "Use of Virtual Environment and Virtual Prototypes in Co-Design: The Case of Hospital Design"

_computers, doi:10.3390/computers8020044_

Round 1

Reviewer 1 Report

This paper describes feedback obtained in a CAVE by hospital workers for various designs of single patient rooms in an intensive care unit.

There are some mayor and minor issues with this work.

Mayor:

The research question “what kind of virtual prototype use situation supports multi-occupational co-design” is not answered systematically. There is a significant amount of research that have been studying the effects of different prototype representations on user preference and behavior. This work only uses a CAVE VR and does not compares with other baseline methods like showing static renderings and or rendered videos of the concept single patient room. This is a mayor flaw in the work.

Minor:

-The hardware and software used in this work needs to be detailed in the manuscript.

-In the background section take into consideration the levels of immersion on virtual reality prototypes, and cite:

de Casenave L, Lugo JE. Effects of Immersion on Virtual Reality Prototype Design Reviews of Mechanical Assemblies. ASME. International Design Engineering Technical Conferences and Computers and Information in Engineering Conference, Volume 7: 30th International Conference on Design Theory and Methodology ():V007T06A044. doi:10.1115/DETC2018-85542.

-Also in the background section Authors are assuming if virtual prototypes are small they are not useful for this research. The authors do not present nor cite other works to support that assumption. 

-There are other ways in which many participants can look at the same VE, the CAVE is not the only option.

-Table 4, there are many single person use VR that can recreate the actual size of a virtual prototype!

-There is a suite of other VR/AR configurations that can do what the authors are doing in this work, the CAVE is not the only way to have many people in a VE for co-design.

- need to add the technical details of the VR set up (software used to build model, software used to build environment, hardware used PCs, projectors, glasses, sensors, etc.) also provide details about the construction of the VE and how it was modified between sessions and how easy it was.

-“ The precision and detail of virtual prototypes increased round by round.” Add more information, add a flow chart of the experiment

-remember that complex systems need high fidelity porotypes, simple products can use low fidelity

-The fourth issue in section four is unclear to the reader.

-in the writing omit fillers like “we learned a lot” (line 330).

-line 348 changing on the fly is good for design but not for a research study.

-line 378 How much expensive? Provide numbers.

-Discussion section: conclusion is weak, this paper is more of a description than a scientific experiment.

In general, the introduction does not provide sufficient background nor references. The method is not adequately described, results are not clear and conclusions are weak at most.

Author Response

We have rewritten many parts of the paper. This is shown in the above file.

Reviewer 2 Report

This paper reports an interesting work on the design of hospital care spaces through a co-design approach in a Virtual Environment (VE) in which hospital staff with different occupations. Participated. The value of the paper is in showing what benefits a co-design approach in a VE can bring to an actual design project. However the paper has a number of weaknesses that need to be addressed. Please find a few suggestions below.

- Positioning and originality of the study: The position of the study in the co-design community could be clarified. In the literature review section, the authors could identify which studies are the closest to their study and highlight the commonalities, differences and their own contribution to the field. The contribution should go beyond data related to the specific design case that has been studied and report generalisable insights that could be useful for a whole research community. For example, what does the study show about CoDesign that could be used in any CoDesign project ?

- Contribution and target audience: Is it not clear what community can benefit the most from the findings. Are the authors targeting CoDesign researchers ? Virtual Reality researchers ? Hospital Designers ? In case all of the aforementioned are the target, then the findings could be structured in a way that highlights different types of findings for different targets. In the current version of the paper, the findings report

- Scientific rigour: The paper has an overall lack of information regarding the research methodology, which makes the validity of the outcomes weak. For example, the following information on the methodological protocol cannot be found in the current paper:

- objective of the protocol: what data were you looking for ? On page 8, there is a mix of findings on the design project outcomes (e.g. size of the hospital rooms) and on the design process (e.g. easiness to communicate around a virtual prototype), so it is not clear what the research focus is.

- protocol and metric: what data were collected during the sessions (e.g. verbal transcripts of conversations, post-hoc subjective comments…) ? in what session ?

- data analysis: how were this data coded, analysed and interpreted ? Section 4 and 5 report findings in a very unstructured way and without supporting data (e.g. page 10 “the speed of the tour was too high”, page 10 “the last prototype was higher fidelity”, page 11 “it is expensive” – no number provided).

- Structure: the structure of the paper could follow the conventional structure of scientific papers, in particular a “results” section seems to be missing. Indeed section 3 (“Case of Hospital Design”) contains information on the methodological approach and it is followed by section 4 (“Benefit 1: Professionals work practice to designer team”) which is an interpretation / discussion of results that are loosely described. In a newly added section on results, the authors could report the transcripts of conversations between participants and their comments on the experience of using the VE. Then

Author Response

(The authors gave the same response as above.)

Reviewer 3 Report

Thank you for this article. I thought this was a very interesting paper but has some needs for improvement.

Methods

Since this reads like a Research through Design style paper, you may want to discuss that in your methods.

I'm not sure (beyond a questionnaire) how the co-design sessions impacted the design iterations. I would need to see more discussion and evidence from the data collected on how that process happened.

Do you have better explanations for the user groups and demographics? I want to see them.

In short, I'm not sure how you collected your data for this study. I would need to see it in detail.

Discussion

"This study demonstrates that virtual prototypes presented in a walk-in VE provides a useful
environment for multi-occupational co-design." This is a bold claim and not one I think you can defend. You can say that *this* prototype helped with *this* group but I'm not sure it is generalizable beyond your study based on what you presented.

Your later discussion points are more fitting for your study.

One other note, the English grammar is this needs a lot of work to be publishable. I was able to discern the points of the study but some of the errors took away from the good work.

Some other notes:

How is line 34 a problem?

Table 1 should be on one page.

Table 2 is hard to read because all text is centered.

Table 4 seems unnecessary

Author Response

(The authors gave the same response as above.)

Round 2

Reviewer 1 Report

No more changes needed.
Computers EISSN 2073-431X Published by MDPI AG, Basel, Switzerland RSS E-Mail Table of Contents Alert
Back to Top